# LLM-Based Code Translation Needs Formal Compositional Reasoning

## ABSTRACT

Recent advances in large language models (LLMs) have achieved impressive performance on source-to-source code translation benchmarks, with potential applications ranging from enterprise code migration to safety-critical software modernization. Yet today's evaluations remain shallow: test suites, syntactic matches, and heuristic similarity metrics conflate superficial success with true reliability. This gap is critical in high-assurance domains, where subtle errors can compromise security, safety, or maintainability. In this position paper, we argue that correctness, not just plausibility, must become the governing principle of LLM-based code translation. Specifically, we argue that principled definitions of correctness, grounded in formal methods and enforced through compositional reasoning, are essential for trustworthy code translation. We propose a layered view of correctness, encompassing top-level functional equivalence, internal contracts and invariants, and non-functional properties such as memory safety and timing guarantees. We highlight why LLMs alone cannot satisfy these obligations, and advocate for hybrid workflows where *formal* reasoning tools constrain, guide, and certify translation both during and after generation, which, we believe, offers a scalable path forward to translating realistic code bases. By embracing compositional specification, translation, and verification, we can turn LLMs from statistical translators into reliable collaborators. Finally, we outline the key open challenges, including cross-language reasoning, specification extraction, and correctness beyond functional equivalence, that must be solved to realize this vision.

## 1 INTRODUCTION

Code translation (or, more generally, code migration) has become a popular benchmark for evaluating the capabilities of large language models (LLMs). Given the development of tooling, datasets, and the rapid progress of LLMs in this domain, code translation has the potential to become widely used in industry. Despite this, the dominant evaluation criteria remain shallow, primarily based around test-case execution, exact match against a ground truth, and occasionally BLEU or Code-BLEU metrics (Ren et al., 2020) for a heuristic notion of similarity. While they serve as proxies that are often convenient to evaluate, these metrics fail to capture the deeper and broader notions of correctness that are required of code used in higher assurance contexts, and when a ground truth solution is not available. Failures of these criteria range from *incomplete coverage*, i.e., the translated program being equal to the source on the tests, while at the same time failing on others, to fundamentally failing to enforce translation guarantees, e.g., the absence of security vulnerabilities, or any standards of maintainability. As a result, current evaluations can conflate superficial success with true reliability, a gap that becomes especially critical in high-consequence domains such as cryptographic libraries, operating systems, or embedded systems, or memory management routines.

We believe it is essential that the AI code generation community looks beyond test-based correctness and considers a broader definition that encompasses *formal reasoning*, including logical specifications and verification. This viewpoint is also supported by recent perspectives on AI safety (e.g., Dalrymple et al. (2024)). We also believe that it is essential that both the formal specification and verification, and the AI-based code translation are *compositional*. That is, the translation task must be broken down into subtasks, and subtasks must have their own specifications. We believe this compositional approach is essential for two primary reasons: scalability of the process and enabling better communication of the intent of the translation.

**Scalability in verification and translation.** Formal reasoning tools typically suffer from scalability issues when applied monolithically to large codebases. Instead, employing compositional verification (McMillan, 1999) by decomposing proof obligations across a set of smaller components enables proof convergence. Decomposition benefits LLM-based translation as well. While LLMs *might* scale to industrial code bases, monolithic translation of large fragments leads to poor accuracy in the best case (Shetty et al., 2024; Zhou et al., 2025), and exceeds context limits in the worst.

**Communication of intent.** Compositional and layered specifications can be easier and more intuitive for some translation tasks; for instance, one may want to define a specification about the inputs to a specific function, or to limit the behavior of a specific datatype. We also hypothesize, based on our experience, that breaking the translation task into smaller compositional tasks will enable more accurate communication of intent to the LLM.

Importantly, we argue it is simply common sense to use compositional translation and verification. The benefits of compositional translation are amply evident, even in code translation based on more conventional, symbolic program synthesis applied to industrial code bases (e.g., (Ahmad et al., 2019)). We see no reason for AI-based code translation to be different in this respect.

There are several challenges to adopting our compositional specification, translation, and verification approach, some of which we outline and attempt to address by means of reference to the literature, and some of which we outline as open challenges for the communities to discuss.

Firstly, defining correctness is the fundamental prerequisite for making progress on code translation. Without an explicit and principled notion of correctness, it is impossible to tell whether improvements on benchmarks reflect genuine advances in reliability or just better performance on selected tests. Yet defining a compositional specification is non-trivial: what counts as a correct translation can depend on language semantics, library behavior, and deployment context. For instance, correctness may mean full semantic equivalence, preservation of safety properties, or adherence to specific invariants depending on the application domain. Establishing such definitions is therefore not just a technical detail but a necessary foundation for trustworthy evaluation and future research.

Once a definition of correctness is reached, the next question is how to enforce it. Automated formal methods (FMs)—theorem provers, software verifiers, equivalence checkers—provide exactly this machinery: they make correctness obligations explicit and machine checkable. Crucially, many of them support compositional verification, allowing correctness guarantees for individual functions or modules to be assembled into proofs of larger systems. Yet traditional automated formal methods have struggled to scale this process to realistic codebases because of the need to *synthesize proof artifacts* such as invariants, abstractions, interface specifications, etc. We argue that progress will come from coupling AI and FM closely: using generative AI not only as a source of candidate translations, but as a generator of proof artifacts that enables compositional reasoning at scale.

**We summarize our proposals and arguments with the following:**
1. Testing alone is insufficient for rigorous evaluation as it is unable to detect subtle but essential correctness criteria. Correctness needs to be assured through *formal* verification.
2. Correctness should be defined as a hierarchy of obligations, encompassing top-level functional equivalence, preservation of internal contracts and invariants, and the maintenance of desirable properties such as safety, security, and resource guarantees.
3. Furthermore, we argue that specification, translation, and verification must be performed *compositionally* to enable scalability to realistic code bases.

## 2 CORRECTNESS: WHY IS IT SUCH A BIG DEAL?

As discussed in Section 1, the current standard for correctness for code translation and generation benchmarks is testing. Benchmarks such as HumanEval (Chen et al., 2021), MBPP (Austin et al., 2021), and SWE-bench (Verified) (Jimenez et al., 2024) evaluate models by whether generated or modified code passes hidden or repository-level test suites, while translation benchmarks, including TransCoder (Roziere et al., 2020), AVATAR (Ahmad et al., 2023), CodeTransOcean (Yan et al., 2023), xCodeEval (Khan et al., 2024), RustRepoTrans (Ou et al., 2024), and TransLibEval (Xue et al., 2025), similarly define success through executable test cases. Beyond evaluation, testing is also being utilized to enhance model performance; self-debugging methods generate or leverage tests to iteratively repair code (Chen et al., 2024), and translation frameworks such as TransCoder-

ST (Roziere et al., 2022) and UniTrans (Yang et al., 2024) regenerate or filter outputs until they pass provided tests. Together, these efforts highlight that testing is considered by the community to be not only the standard for evaluating correctness but also a de facto *definition of correctness*. We will now explain why we believe this is insufficient.

## 2.1 WHEN AND WHY DOES TEST-BASED CORRECTNESS FAIL?

**Reason 1: Testing cannot capture full equivalence.** Even simple translation tasks, such as Figure 1 (from Cheung et al. (2013) on verified code translation of web applications) illustrates the flaws in a testing-only approach to correctness.

The source program iterates over each record r and returns adults under 65 years old, and the goal is to improve performance by removing the explicit loop. The code translator initially produced the incorrect translation out = [], which *passes all the generated tests* as the tests only enumerate ages $\in [0, 2^3]$. On these test inputs, both the source and translated programs return empty lists. Even after increasing the test interval to $[0, 2^6]$, the newly produced (incorrect) translation (shown in translation.py) still passes all tests, as all test ages are $< 65$.

```
                                    source.py
out = []
for r in table:
  if r.age > 21 and r.age < 65:
    out.append(r)
```

```
                                translation.py
out = table.filter(lambda r: r.
    age > 21)
```

Figure 1: Pitfalls of testing: source input (top), and incorrect translation (below). Translated code misses the upper bound for the age filter.

**Reason 2: Testing alone cannot capture adequate context, especially for deep specifications.** Almost all of the previously discussed prior work uses tests to capture top-level input-output equivalence. Many correctness obligations, however, rely on *deep specifications*, i.e., behaviors of non-top-level functions/APIs such as memory layouts, build configurations, or hardware protocols. In such cases, a test-based input-output specification is fundamentally insufficient to both guide generation (e.g., through in-context examples) and to evaluate it.

***Example.*** *Figure 2 contrasts the project-level ownership convention with translations generated without that context. In the C code,* init_consumer *transfers ownership of the buffer to the callee, so the caller must not free it, and* shutdown_consumer *later reclaims it. A naive Rust translation that only borrows the* Vec<u8> *leaves the global dangling once the caller drops it, making* shutdown_consumer *impossible to implement without redesign.*

```
                                    consumer.c
// buf.h
struct buf { uint8_t *data; size_t cap; };
struct buf *alloc_buf(size_t cap);
void free_buf(struct buf *b);

// consumer.c (snippet shown to translator)
static struct buf *g_buf;

void init_consumer(struct buf *b) {
    g_buf = b;  // takes ownership; caller
    must not free 'b' afterwards
}
```

```
                                        buf.rs
// buf.rs (generated without seeing ownership
    convention)
pub struct Buf<'a> { pub data: &'a mut Vec<u8
    > }

static mut G_BUF: Option<&'_ mut Vec<u8>> =
    None;

pub fn init_consumer(buf: &mut Vec<u8>) {
    unsafe { G_BUF = Some(buf); } // borrow
    only
}
```

```
                            shutdown_consumer.c
// shutdown_consumer.c (translated later,
    relies on ownership transfer)
extern struct buf *g_buf;
void shutdown_consumer(void) {
    free_buf(g_buf);
    g_buf = NULL;
}
```

```
                            shutdown_consumer.rs
// downstream translation now clashes with
    the borrowed global
unsafe pub fn shutdown_consumer() {
    if let Some(buf) = G_BUF.take() {
        free_buf(buf); // type error!
    }
}
```

Figure 2: Ownership mismatch in C-to-Rust translation. C transfers ownership, but Rust only borrows the buffer, leaving a dangling reference.

**Reason 3: Tests alone cannot capture challenging semantic requirements.** Even when the entire source fragment is present in the prompt, an LLM can violate correctness because it lacks understanding of the semantics and/or the notion of correctness that needs to be captured. Limited unit tests rarely expose these challenging notions of semantic correctness during testing or during training, resulting in poor performance by LLMs when evaluated against these requirements.

```
                                            hash.java                                                hash.c
int hash(byte[] s) {                                      int hash(const uint8_t *s, size_t n) {
  int h = 0;                                                int h = 0;
  for (byte b : s) {                                        for (size_t i = 0; i < n; i++) {
    // May overflow, but defined wraparound                  // signed overflow is UB in C!
    h = 31 * h + (b & 0xff);                                 h = 31 * h + s[i];
  }                                                         }
  return h;                                                 return h;
}                                                         }
```

Figure 3: Integer overflow mismatch in Java-to-C translation. Naively using `int` in C violates Java's well-defined wraparound semantics, introducing undefined behavior.

```
                                            verify.py                                                verify.c
import hmac                                                #include <string.h>
                                                          #include <stdbool.h>
def verify(token, secret):
    # constant time                                       bool verify(const char *token, const char *secret) {
    return hmac.compare_digest(                               return strcmp(token, secret) == 0; // exits early
        token, secret)                                    }
```

Figure 4: A key verification function in Python (left) translated to C (right). The translated code introduces a side channel that will not be detected without an explicit constraint.

1. *Language semantics.* LLMs can violate top-level functional I/O or internal contract equivalence due to semantic differences in language constructs such as type mapping, undefined behavior, or operation semantics (Shetty et al., 2024; Zhou et al., 2025).

   ***Example.*** *Consider the Java to C translation in Figure 3. In Java,* `int` *arithmetic is defined modulo* $2^{32}$*; the naive C uses* signed `int`, *where overflow is* undefined behavior. *Compilers may assume "no signed overflow", leading to compilation succeeding, but a divergence in the behavior of the hash function. A correct translation would implement explicit mod-$2^{32}$ arithmetic (e.g.,* `uint32_t` *or 64-bit + mask) and only reinterpret the low 32-bits at the end.*

2. *Broader safety, security, and resource properties.* Real-world software often relies on properties beyond pure input-output functional behavior, such as memory safety and resource usage (Gong et al., 2025; Farrukh et al., 2025). Even with the entire source provided in-context, LLMs may lack semantic understanding of such properties, resulting in the translated code not adhering to the same requirements as the source.

   ***Example.*** *In Figure 4, Python's standard library exposes* `hmac.compare_digest`, *which guarantees data-independent timing. When asked to "translate this module to C" with only the function body in view, an LLM-backed transpiler routinely emits code similar to the second code block in Figure 4. This code is wrong because, whilst the C code is functionally equivalent,* `strcmp` *short-circuits at the first differing byte, reintroducing an observable timing side channel. Source projects often call high-level constant-time helpers like this, whose semantics are only described in documentation.*

**Reason 4: Compositional translation with testing alone is prone to backtracking.** Larger translation tasks aggravate all of the problems above. Syzygy (Shetty et al., 2024) performs an LLM-driven translation of `zopfli` (Google, 2024). They perform a compositional (function-by-function) translation of the codebase in dependency order. Even with a large number of tests (their verification approach), they run into failure on a function later in dependency order due to a previous function (`zopfli_block_split_lz77`) being incorrectly translated (ref. Shetty et al. (2024), Figure 8). While they were able to perform a small, local (and manual) fix, such remediation approaches are not scalable. In general, in the context of compositional translation, late detection might require expensive *catastrophic backtracking*, i.e., a global repair of several previously translated functions.

All of these failure modes are due to test-based specifications capturing an insufficient amount of information. We can categorize these by considering the level of the information that is missing: is it information about *top-level functional equivalence*, i.e., whether the program produces the same output for every input; or is it information about *internal equivalence* of specific components of the code, for example, providing a loop invariant; or is the information missing something *beyond functional equivalence*, such as resource usage.

## 2.2 Why is Defining Correctness Hard?

Even though tests suffer from the previously discussed inadequacies, we tend towards test-based specifications because defining correctness is hard! The key issue is that correctness in code translation is never absolute: while *anchored* in the properties of the source program, it must still be *contextualized to the overall objective* for which the translation is being performed.

**Correctness is contextual.** For example, in certain code translation contexts (e.g., educational/prototyping tasks), ensuring only the top-level input/output (I/O) equivalence (that most test-based approaches aim to capture) is adequate. However, in other contexts, one might additionally desire more aspects of $s$ to be preserved in $t$ (e.g., API boundaries, OS interactions). In this sense, correctness is a *boundary-setting exercise*, specifying which properties must be preserved and where deviations are acceptable. Thus, defining correctness requires communicating *intent* between the translation and downstream application development (more examples are in Table 2).

**LLMs cannot embody an intrinsic notion of correctness.** LLMs approach translation as conditional generation of target program $t$ given the source program $s$ and an (optional) prompt $p$, and parameterized by $\theta$: $M_\theta(t|s, p)$. Translation intent can be provided in two places: first in the model parameters $\theta$ (during training) and second in the prompt $p$ (at inference time). Both of these merely strengthen statistical mappings between certain $s$ and $t$ while weakening others. Thus, unlike classical PL-based techniques, e.g., rule-based translators, that have correctness explicitly noted via formal notions[1], "correctness" for LLMs is implicit in these mappings derived from training examples. However, correctness cannot be specified, let alone guaranteed, by likelihood alone, making an explicit formal definition imperative.

## 2.3 A layered view of correctness

Our position is that correctness is a *spectrum of obligations*, which must be formalized and then can be checked at different levels of strength. We now provide a high-level overview of this spectrum, based on the failure modes observed before[2].

**Top-level functional equivalence.**

Top-level functional equivalence requires that, for any possible set of inputs $x$, the two pieces of code produce the same output. That is, $[\![s]\!](x) = [\![t]\!](x)$, where $[\![\cdot]\!]$ denotes program semantics, including return values and externally visible side effects.

As discussed previously, testing, or input-output (I/O) equivalence, is the default correctness metric used by the community. In top-level I/O equivalence, the list of possible inputs for which $[\![s]\!](x) = [\![t]\!](x)$ must be maintained is limited to a finite list of inputs (either selected randomly, or given by the user). It is infeasible to guarantee full top-level functional equivalence (i.e., equivalence for any value of $x$) with testing since this would require testing on a number of inputs so large as to be practically infinite. Beyond testing, one can look to formal methods such as model checking (Clarke et al., 2003) to prove full top-level functional equivalence, but the problem is undecidable. As a result, full monolithic proofs are often infeasible without significant amounts of human effort.

***Example.*** *Consider the* `node.h` *and* `block.c` *example files given in Figure 5. If the* `operate_on_data` *function is the top-level (user-facing) API exposed by this code, we would only require that an I/O specification of the form* $\phi$(`datas`, `result`) *be preserved on the target codebase, permitting internals of the C code, e.g., the* `struct` *definition, to be modified/optimized.*

**Internal equivalence.** Below top-level functional equivalence, there are many other forms of alignment between $s$ and $t$ that we may wish to describe. For instance, at a function level, providing function contracts in the style of Hoare logic (Hoare, 1969) allows us to provide logical predicates about the behavior of the translated code (or functions within it) and assumptions under which those predicates must hold. Formally, if the source program $s$ satisfies a Hoare triple $\{P\}\ s\ \{Q\}$, then the

---

[1]Also known as "correct by construction."

[2]Syntactic validity (e.g., producing compilable, well-typed code) is a generic prerequisite for code generation and has been addressed via rejection sampling (Dou et al., 2024), constrained decoding (Tromble & Eisner, 2006; Beurer-Kellner et al., 2024; Park et al., 2024), and hybrid symbolic search (Barke et al., 2024; Li et al., 2024b; 2025); our focus is on richer correctness obligations beyond this baseline.

```
                                                node.h                                               block.c
struct node {                                            node* create_block (data_t * datas) {
  node * second; data_t * data;                            node* fst = (node *) malloc(sizeof(node));
} node;                                                    node* snd = (node *) malloc(sizeof(node));
                                                           snd->second = NULL;
                                                           fst->second = snd;
                                              anyblock.c   // ... populate data
node* create_block_by_size (                               return fst;
  data_t* datas, int cnt) {                              }
  node* curr = (node*) malloc(sizeof(node));
  node* next = NULL;
  for (int i=0; i < cnt; i++) {                          result_t operate_on_block (node* block) {
    curr->second = next;                                   // ... some operation on a block
    // ... populate data                                 }
    next = curr;
    curr = (node *) malloc(sizeof(node));                result_t operate_on_data(data_t* datas) {
  }                                                        node* block = create_block(datas);
  return curr;                                             result_t result = operate_on_block(block);
}                                                          return result;
                                                         }
```

Figure 5: Examples of top-level vs. internal equivalence. Preserving only I/O equivalence allows structural optimizations, but future extensions (e.g., variable-length blocks) may break.

translated program $t$ must also satisfy $\{P\}\ t\ \{Q\}$. These contracts can be extended to reason about concurrency properties (O'Hearn, 2004).

For programs with loops, we can also consider loop invariants. If $s$ contains a loop, for which $I$ is a valid loop invariant (i.e., $I$ holds on entry to the loop, and after every execution of the loop), then we require that the translated loop preserves the same invariant.

Contracts and invariants are used to verify LLM-based formal model generation by Misu et al. (2024), and in source code translation by Bhatia et al. (2024). LLMs are even used to provide the function contracts in Sun et al. (2024), where they are auto-formalized from doc strings, and in Chen et al. (2025), where LLM-provided contracts are used to enable verification of polyglot systems. A natural bonus of using contract equivalence is that it enables us to break down a verification problem into subproblems and deploy scalable compositional verification approaches.

***Example***. *Referring back to Figure 5, if we only prescribed top-level functional equivalence, the translation might leverage the fact that* block.c *only creates length 2 node-chains, and may choose to flatten a chain of two nodes into a single two-data element* node *struct. However, one could later add a* create_block_by_size *function that creates an arbitrary-length chain of blocks. A migration of this function would not be supported by the earlier translation that only ensures top-level I/O equivalence. A translation that preserves* structs *exactly would help in this.*

*Previous work takes ad-hoc approaches to navigate the boundary of internal equivalence. CRUST-Bench (ref. Khatry et al. (2025), Figure 1) requires Rust struct* impl *signatures to be manually specified, while Syzygy Shetty et al. (2024) requires the same for struct fields.*

**Beyond functional equivalence.** We may also require that the translated program preserves equivalence w.r.t. lower-level aspects of program execution, such as ABI conventions, time/space complexity, memory accesses, and system calls performed. Prior work in program synthesis has incorporated runtime and memory footprints to guide search (Collie & O'Boyle, 2021; Hu et al., 2021), while work on program repair has used security properties as correctness criteria (Tihanyi et al., 2025). Such obligations remain underexplored in the context of LLM-based code translation.

***Example***. *As an illustrative example, for Figure 5, one may require that the data layout of the* node *struct be preserved between the source (C) and the translated programs. As a more realistic example, Li et al. (2024a) discusses this challenge in the context of the Rust-for-Linux project: "The major difficulty of writing safe drivers in Rust is to reconcile the inflexibility of Rust versus kernel programming conventions ..." (ref. Li et al. (2024a), Sec. 3).*

**Takeaway.** This list of correctness levels is not exhaustive, and many other possible specification types exist (and those that do not exist in the literature yet, may do in the future). As a result of these many levels, we believe correctness in translation is best viewed as an *explicit, layered constraint set* that depends both on the migration goal and on the available verification budget; it is not a byproduct of translation but the *lens* through which translation quality must be defined and enforced. This constraint set serves two roles: (i) it defines validity for a candidate translation, and (ii) it provides the feedback signal for online (during generation) or offline (after generation) compositional verification.

# 3 Achieving Correctness: Why LLMs Alone Fall Short, and Candidate Enforcement Strategies

| When | Approach | Top-level I/O | Internal | Beyond Functional | Domain | Representative Work |
|---|---|---|---|---|---|---|
| During | Grammar-constrained decoding | ○ | ○ | ○ | Code Generation | Yin & Neubig (2017) |
| | Type- and scope-aware generation | ○ | ○ | ○ | Code Generation | Mündler et al. (2025) |
| | Execution-based validation | ◑ | ○ | ○ | Code Generation | Lavon et al. (2025) |
| | Intermediate contract/test checking | ● | ○[2] | ○ | Code Translation | Zhou et al. (2025) |
| | Test generation and repair | ● | ◑ | ○ | Code Translation | Gu et al. (2024) |
| | Verifier-in-the-loop | ● | ● | ●[1] | Program Synthesis | Kalyan et al. (2018) |
| | Constraint-solver guided | ● | ● | ●[1] | Program Synthesis | Zhang et al. (2018) |
| After | Learning-based execution filtering | ◑ | ○ | ○ | Code Generation | Ni et al. (2023) |
| | End-to-end test filtering | ● | ◑ | ○ | Code Translation | Farrukh et al. (2025) |
| | Verified transpilation | ● | ● | ●[1] | Code Translation | Bhatia et al. (2024) |

[1] Assumes the property is explicitly modeled and compositionally enforced (incl. beyond-functional, e.g., constant-time, information-flow); otherwise mark as ◑.

[2] If checks include contracts or properties that expose and validate internal invariants, interpret as ◑.

Table 1: Coverage of verification mechanisms across correctness layers (top-level I/O, internal contracts, beyond-functional), grouped by timing (During vs. After generation). ● means can guarantee; ◑ means contributes useful evidence; ○ means not helpful.

As stated in Section 2.2, LLMs are powerful sequence models but not correctness engines. They optimize local likelihood, not global semantics, and thus fail to guarantee the layered definitions of correctness described in Section 2.3. For each layer, we outline the key shortcomings and the enforcement strategy (that can occur during or after generation) that can compensate.

**Top-level functional equivalence.** *Why LLMs fail?* Passing finite I/O tests is within reach of today's models, but ensuring $[\![s]\!](x) = [\![t]\!](x)$ for all $x$ is infeasible; LLMs generalize poorly beyond observed inputs and default to surface similarity rather than semantic preservation. This explains why testing dominates evaluation in practice (Ahmad et al., 2023; Khan et al., 2024; Xue et al., 2025), despite its incompleteness.

*Enforcement:* Online checks such as grammar- and type-constrained decoding ensure every prefix is syntactically and type valid (Yin & Neubig, 2017; Mündler et al., 2025), while offline execution filters and large I/O test suites (Ni et al., 2023; Farrukh et al., 2025) increase behavioral coverage. Formal verification of top-level functional equivalence, e.g., via model checking Clarke et al. (2003), suffers from scalability issues, but bounded proofs of functional equivalence are feasible (Brauckmann et al., 2023). Furthermore, LLMs themselves can be used to produce proof artifacts to enable verification to scale, via compositional verification (Chen et al., 2025) and invariant generation (Pirzada et al., 2024). Verification outcomes can be recycled into reinforcement learning loops (Jha et al., 2025), nudging models toward functional equivalence beyond finite test sets.

**Internal equivalence.** *Why LLMs fail?* Models often "optimize away" internal structures that seem redundant locally (e.g., simplifying `struct`s or omitting invariants), thereby breaking contracts and future extensibility while still passing top-level tests. Benchmarks such as CRUST-Bench (Khatry et al., 2025) and systems like Syzygy (Shetty et al., 2024) and SACTOR (Zhou et al., 2025) illustrate how brittle these internal interfaces are in practice.

*Enforcement:* Online strategies such as contract or test checking (Zhou et al., 2025; Shetty et al., 2024), and verifier-in-the-loop pruning with symbolic execution or SMT solvers (Kalyan et al., 2018), prevent partial outputs from violating local specifications. Offline compositional veri-

fiers (Bhatia et al., 2024) then certify function- or module-level contracts, assembling scalable global proofs from locally validated pieces.

**Beyond functional equivalence.** *Why LLMs fail?* Non-functional and security obligations, such as ABI compatibility, resource bounds,and memory safety are rarely captured in training corpora and have no proxy in next-token likelihood. As a result, LLMs may silently introduce regressions in efficiency or safety. This is especially visible in safety-critical domains such as drivers, where reconciling Rust's safety model with Linux kernel conventions remains a major blocker (Li et al., 2024a).

*Enforcement:* Constraint-guided decoding (Zhang et al., 2018), type- and scope-aware pruning (Mündler et al., 2025), or lightweight static analyzers can act online to prevent unsafe continuations. Similarly, static analyzers have been used to detect security flaws and guide LLMs to repair said flaws in their own code (Tihanyi et al., 2025). Offline analyzers and monitors (Collie & O'Boyle, 2021; Hu et al., 2021) have been used in enumerative program synthesis to certify global properties such as timing budgets, resource usage, but are still relatively under-explored in the world of LLM-generated code.

**Note:** Details about each of these enforcement approaches can be found in Table 1 and Appendix B.

> **Takeaway.** LLMs alone cannot satisfy layered correctness definitions: they overfit to surface plausibility, overlook internal structure, and ignore non-functional obligations. Achieving trustworthy translation requires a *hybrid verification loop*. Online verification prunes infeasible prefixes and enforces local obligations at the level of syntax, types, and function contracts, ensuring that components are correct before composition. Offline verification then certifies global invariants and discharges whole-program obligations, completing the compositional proof. Recycling both forms of feedback into training is essential: without this hybrid, correctness breaks down into brittle heuristics rather than principled guarantees.

## 4 OPEN PROBLEMS

We now describe a number of research challenges inspired by our vision. A recurring theme across these challenges is a fundamental tension: are the bottlenecks due to the *absence of sufficiently expressive specifications*, or due to the *limitations of current verification techniques* in enforcing them? In practice, both obstacles arise: sometimes the difficulty lies in articulating correctness properties (e.g., idiomaticity, timing security), and sometimes in scaling verification across languages, partial programs, or semi-structured standards. Understanding where the barrier lies is itself a research problem, and it shapes how progress can be made toward correctness-aware translation.

**Porting Verification to Code Translation.** Verification-guided generation has shown promise in program synthesis and code generation tasks, but adapting these techniques to the *code translation* setting remains an open problem. Unlike synthesis, where the goal is to produce *any* program satisfying a specification, translation requires preserving the semantics of a *specific* source program across languages. This raises unique difficulties: (i) partial prefixes in the target may not admit a verifier interface until a large unit (e.g., a function or module) is complete, limiting the granularity of online checks; (ii) semantic preservation often involves global invariants (memory safety, resource usage, security properties) that do not decompose cleanly into local contracts; and (iii) formal equivalence checking requires reasoning about the semantics of both source and target languages, as well as bridging across type systems and memory models. Existing verifiers and test suites are almost always tied to a single language, making cross-language correctness harder to establish. Thus, the bottleneck is twofold: we lack sufficiently expressive *specifications* of semantic preservation across languages, and we also lack scalable *verification* techniques to check these properties compositionally during and after translation. Designing modular equivalence checkers that can span heterogeneous toolchains is therefore a central open challenge.

**Checking Security Properties Beyond Semantic Equivalence.** Some critical applications require correctness criteria that go beyond functional equivalence, especially in domains like cryptography and systems security. For example, constant-time execution is essential for preventing timing side-channel leaks in cryptographic libraries, yet this property is not implied by functional correctness. A translation may preserve input–output semantics but inadvertently introduce data-dependent

branches or memory accesses, violating security requirements. Prior work has shown how properties such as constant-time execution, non-interference, and controlled information flow can be formally specified (Kozyri et al., 2022; Lee et al., 2022), but the challenge is twofold: (i) specifications are often highly domain-specific, requiring specialized logics or refinement types; and (ii) verification tools for these properties are typically language-specific and do not port across source and target. Thus, the open problem is not just to verify security properties, but also to determine whether they can be integrated into a *general correctness specification* for translation, one that can be stated once, and checked across diverse languages and compiler toolchains.

**Translation for Code Modernization and Maintenance.** Beyond correctness, practical translation often serves a modernization goal: making legacy code more maintainable, extensible, and accessible to new developers. For instance, the fish shell project recently migrated from C++ to Rust to improve ergonomics and lower the barrier for contributions[3]. If LLMs assist such efforts, semantic preservation is necessary but not sufficient: the translated code must also be *idiomatic* in the target language, leveraging its abstractions, patterns, and ecosystem. Otherwise, the result may be correct but unreadable, brittle, or "foreign" to the target community. This raises a specification challenge (how do we formally describe what makes code idiomatic or maintainable?) and a verification challenge (how do we check that LLM outputs meet such "soft" criteria?). Here, correctness must be expanded to include human-centered notions of quality, suggesting a hybrid verification regime that mixes formal contracts with automated style, idiomaticity, and maintainability checks.

**Specifications Given in Semi-Structured Form.** In many domains, the only available specifications are in semi-structured natural language or standards documents, rather than in machine-checkable logics. For example, the IETF publishes protocol specifications in semi-formal text, such as the one accompanying Google's recent open-source zero-knowledge proof library in C++[4]. Translating this code into Rust or another language while maintaining compliance with the IETF standard requires bridging between the semi-structured specification and a formal logic suitable for verification. This raises both a specification and verification gap: extracting precise logical obligations from semi-structured text, and then developing verifiers that can check them in the target code. Automating this pipeline, perhaps by leveraging LLMs to synthesize formal contracts from standards documents, remains an open and underexplored challenge.

**Verification of Partially Translated Codebases.** Real-world translation rarely happens all at once; instead, codebases are often partially translated, yielding polyglot systems where components in different languages must interoperate. This complicates verification: most state-of-the-art tools are specialized for a single language and cannot reason across boundaries. Preliminary work by Chen et al. (2025) shows how LLMs can synthesize interface contracts to connect different verifiers, enabling compositional reasoning across languages. Yet several challenges remain: (i) ensuring that cross-language contracts are both sound and precise enough for verification; (ii) scaling compositional reasoning to large, evolving codebases; and (iii) integrating heterogeneous toolchains with different logics and proof obligations. Here, the bottleneck is primarily verification, but specification plays a role too: we need principled ways to describe cross-language invariants, not just within each isolated component.

## 5 CONCLUSION

LLMs can translate syntax, but they do not preserve semantics. Treating surface similarity as success risks brittle systems that fail when correctness truly matters. In this position paper, we argue that the future of code translation is not in building larger models validated by test sets but in establishing formal notions of correctness as the governing principle. We show how formal methods can be used as a means to define correctness, and propose different mechanisms to utilize formal methods in code translation. We then discuss how compositionality can scale code translations, and discuss emergent challenges where we envision similar techniques can be applied.

---

[3]https://fishshell.com/blog/rustport/
[4]https://github.com/google/longfellow-zk

## ETHICS STATEMENT

This paper does not introduce new datasets, models, or systems; instead, it presents a position and research agenda. All examples and case studies are drawn from publicly available sources or prior work, with appropriate citations. No sensitive or personally identifiable information is used. Our arguments highlight the importance of correctness and assurance in code translation, especially for high-consequence domains, precisely to mitigate downstream ethical risks of deploying unreliable translations.

**Human Oversight and Accountability.** The authors accept full responsibility for every scientific claim and for the correctness of all results presented in the paper. No proprietary or confidential data were provided to external services during the preparation of this work.

## REPRODUCIBILITY STATEMENT

For reproducibility, we have provided all conceptual taxonomies, definitions, and comparisons in a way that is self-contained and directly tied to prior literature. Where we discuss benchmarks, tools, or verification protocols, we reference publicly available resources to enable others to trace and replicate the reasoning. Future empirical work following this agenda should adhere to principles of transparency, dataset documentation, and open release of code and evaluation frameworks to foster cumulative progress. We present no new code or experiments for this paper.

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

## A    APPLICATION DOMAINS

| Application Domain | Top-level I/O | Internal | Beyond Functional | Representative Benchmark |
|---|---|---|---|---|
| Educational / prototyping | ◑ | ○ | ○ | Chen et al. (2021) |
| Enterprise code migration | ● | ◑ | ○ | Jimenez et al. (2024) |
| API migration | ● | ◑ | ○ | Xue et al. (2025) |
| C → Rust memory safety | ● | ● | ◑ | Khatry et al. (2025) |
| Safety-critical systems | ● | ● | ● | Chang et al. (2025) |
| Cryptographic implementations | ● | ● | ● | Cai et al. (2024) |

Table 2: Correctness notions emphasized across different application domains. Some of the representative benchmarks come from code generation area, but they share the same notion of correctness. ● means required guarantee; ◑ means typical evidence but short of a guarantee; ○ means usually unnecessary.

## B    ACHIEVING CORRECTNESS: ONLINE & OFFLINE APPROACHES

Given a sufficient layered correctness specification, the question is how can we enforce this? In this section, we examine how existing verification levers can be arranged to support compositional translation with a layered spectrum of specifications. We analyze the verification machinery through the lens of compositional translation, asking how each mechanism supports breaking the task into smaller, spec-annotated fragments whose local guarantees compose. In practice there are two complementary paradigms for enforcing correctness specifications $\mathcal{C}(s)$: constraining or guiding generation *during* decoding (Shetty et al., 2024; Mündler et al., 2025; Zhou et al., 2025), or certifying candidates *after* they are produced (Ni et al., 2023; Bhatia et al., 2024; Farrukh et al., 2025). We interpret these paradigms as design points for injecting formal signals that keep the translation process compositional, and we compare them in terms of how effectively they preserve and exploit modular structure.

### B.1    ONLINE VERIFICATION: VERIFICATION DURING GENERATION

Verification during generation integrates correctness checks directly into the decoding process. Rather than treating generation and verification as separate stages, the model is guided at intermediate steps toward outputs that remain consistent with $\mathcal{C}(s)$. The central idea is that correctness can act as a *search prior*: pruning infeasible partial hypotheses and allocating resources only to viable continuations while maintaining the compositional scaffold introduced in Section 1. Local checks on prefixes or completed subprograms operationalize the idea that each component should carry its own specification.

Formally, translation can be viewed as a sequential decision process $t = (a_1, \ldots, a_n)$, where each $a_i$ denotes an atomic generation decision (e.g., emitting a token or expanding a grammar production like a function, an expression, etc.). For any prefix $a_{1:k}$ we ask whether it admits a *correct extension*: there must exist some completion $t' = (a'_1, \ldots, a'_n)$ with $a'_i = a_i$ for all $i \leq k$ such that $t'$ satisfies $\mathcal{C}(s)$. We encode this requirement with a viability predicate

$$V_{\mathcal{C}}(a_{1:k}, s) = \begin{cases} 1 & \text{if } \exists t' \models \mathcal{C}(s) \text{ with prefix } a_{1:k}, \\ 0 & \text{otherwise.} \end{cases}$$

Only prefixes with $V_{\mathcal{C}}(a_{1:k}, s) = 1$ remain in the search frontier, ensuring that partial hypotheses keep the potential to assemble into a specification-respecting translation.

**How to Enforce It.**    In practice, the predicate can be approximated by layering increasingly strong checks (syntax, types, contracts, property tests, security rules) as computational resources permit. Several strategies have been explored for embedding correctness checks directly into decoding. While most prior work applies these techniques in *program synthesis* or *code generation*, rather than in translation, they illustrate the design space of mechanisms that could transfer to the translation setting. They vary in strength of equivalence guarantees, as summarized in Table 3.

*1. Grammar-constrained decoding.* Decoding can be restricted so that every prefix forms a valid AST or parse tree (Yin & Neubig, 2017). This ensures syntactic well-formedness but does not

capture semantic behavior. More formally: $V_{\mathcal{C}}(a_{1:k}, s) = 1$ iff the prefix parses to a valid AST fragment.

*2. Type- and scope-aware generation.* Lightweight type checkers can prune partial outputs that violate typing rules or variable scope (Mündler et al., 2025). This enforces local well-formedness; full semantic equivalence is not guaranteed. More formally: $V_{\mathcal{C}}(a_{1:k}, s) = 1$ iff no type or scope violations are present in the partial AST.

*3. Execution-based validation.* During generation, partial snippets (e.g., the current line, block, or function) are executed to steer decoding; candidates that violate expected behavior are pruned (Lavon et al., 2025). *More formally:* for a completed subprogram $u \subseteq t$ (or a runnable prefix with a harness), $V_{\mathcal{C}}(u, s) = 1$ iff executing $u$ under its local harness does not crash and satisfies the local semantic checks attached to $u$.

*4. Intermediate contract or test checking.* Once a function or module is complete, it can be validated against unit tests or pre/post-conditions (Zhou et al., 2025; Shetty et al., 2024). Passing such checks implies behavioral equivalence with respect to the chosen test suite, though completeness depends on specification coverage. More formally: for a completed component[5] $u \subseteq t$, $V_{\mathcal{C}}(u, s) = 1$ iff $u$ satisfies the specification (contracts or tests) associated with $s$.

*5. Test generation and repair.* Rather than relying on a fixed suite, additional tests can be *generated* or existing ones *repaired* to expand behavioral coverage and expose corner cases that would otherwise remain untested (Gu et al., 2024; Schäfer et al., 2023; Hu et al., 2025). More formally: $V_{\mathcal{C}}(t, s) = 1$ iff $t$ passes the adaptively grown suite $\mathcal{U}_s^{(\text{iter})}$, where $\mathcal{U}_s^{(\text{iter})}$ includes newly generated or repaired tests until convergence.

*6. Verifier-in-the-loop.* Symbolic execution, SMT solving, or interpreters can be invoked mid-generation to prune infeasible continuations (Kalyan et al., 2018). This offers stronger local semantic guarantees than tests alone, but remains constrained by verifier scalability (e.g., path explosion, solver timeouts). More formally: $V_{\mathcal{C}}(a_{1:k}, s) = 1$ iff the verifier does not refute feasibility of $a_{1:k}$.

*7. Constraint-solver guided.* Here, a formal method such as a constraint logic programming system (e.g., miniKanren) defines the search space itself, while the neural model guides exploration (Zhang et al., 2018). Unlike verifier-in-the-loop pruning, correctness constraints are *always enforced*, yielding provably consistent completions when constraints are satisfiable. More formally: the feasible set is $\mathcal{T}_{\mathcal{C}}(s) = \{t \in \mathcal{T} : t \models \mathcal{C}(s)\}$, and generation is restricted so that at every step $k$, $V_{\mathcal{C}}(a_{1:k}, s) = 1$ iff there exists some $t \in \mathcal{T}_{\mathcal{C}}(s)$ with prefix $a_{1:k}$.

| Approach | Top-level I/O | Internal | Beyond Functional | Domain | Representative Work |
|---|:---:|:---:|:---:|:---:|:---:|
| Grammar-constrained decoding | ○ | ○ | ○ | Code Generation | Yin & Neubig (2017) |
| Type- and scope-aware generation | ○ | ○ | ○ | Code Generation | Mündler et al. (2025) |
| Execution-based validation | ◑ | ○ | ○ | Code Generation | Lavon et al. (2025) |
| Intermediate contract/test checking | ● | ○[2] | ○ | Code Translation | Zhou et al. (2025) |
| Test generation and repair | ● | ◑ | ○ | Code Translation | Gu et al. (2024) |
| Verifier-in-the-loop | ● | ● | ●[1] | Program Synthesis | Kalyan et al. (2018) |
| Constraint-solver guided | ● | ● | ●[1] | Program Synthesis | Zhang et al. (2018) |

[1] Assume the property is explicitly modeled and compositionally enforced (incl. beyond-functional, e.g., constant-time, information-flow); otherwise mark as ◑.

[2] If the checks include contracts or properties that expose and validate internal invariants, interpret as ◑.

Table 3: Coverage of verification-during-generation mechanisms across three correctness layers (top-level I/O, internal contracts, beyond-functional). ● means can guarantee; ◑ means contributes useful evidence; ○ means not helpful.

### B.2 OFFLINE VERIFICATION: VERIFICATION AFTER GENERATION

Verification after generation treats correctness as a filter over completed candidates. The model proposes a set of translations, and verification then selects the subset satisfying $\mathcal{C}(s)$. This *generate-and-verify* approach is conceptually simpler, requiring no modification of decoding, but shifts all validation costs to the end. It is worth noting that verification after generation can be directly in-

---

[5]A *completed component* $u \subseteq t$ is a syntactically closed subprogram (e.g., a function, class, or module) generated during translation, such that $u$ admits standalone semantic checks (contracts, unit tests, or type rules).

tegrated into training via a reinforcement learning loop, turning post-hoc checks into optimization signals (Jha et al., 2025). Concretely, the model samples $k$ candidates $t^{(1..k)} \sim M_\theta(\cdot \mid s)$; a verifier executes tests or invokes formal checks to obtain pass/fail (or graded) feedback; these outcomes are mapped to rewards $r(t^{(i)}, s)$ that drive policy-gradient or actor-critic updates of $M_\theta$. This "generate $\rightarrow$ verify $\rightarrow$ reinforce" loop increases probability mass on specification-satisfying candidates and downweights failure modes. In code region, unit-test/compilation signals have been used as rewards in actor-critic/PPO fine-tuning, improving functional correctness (Le et al., 2022; Shojaee et al., 2023). Beyond tests, deduction and counterexample-guided verification provide richer reward shaping for program synthesis (Chen et al., 2020). Although many examples come from code generation, the same principle applies to *translation*. For example, CoTran fine-tunes a translator using compiler and symbolic-execution feedback as rewards to improve compilation and semantic equivalence (Jana et al., 2023).

Although conceptually simpler than online verification, offline verification plays a crucial *compositional closing role*: it aggregates the guarantees established during generation and certifies that the fully assembled program satisfies the remaining whole-program obligations (e.g., integration invariants, security policies, performance budgets). This stage is what turns a set of locally verified components into a single, trustworthy system.

Mathematically, given a finite candidate set $\mathcal{T}_k(s) = \{t^{(1)}, \ldots, t^{(k)}\}$ sampled from $M_\theta(\cdot \mid s)$, verification acts as a projection defined by a verifier $V_\mathcal{C} : \mathcal{T} \times \mathcal{S} \rightarrow \{0, 1\}$:

$$\hat{\mathcal{T}}(s) = \{t \in \mathcal{T}_k(s) : V_\mathcal{C}(t, s) = 1\}.$$

**How to Enforce It.** Offline verification is implemented by running checkers over fully generated candidates and retaining only those that satisfy $\mathcal{C}(s)$. Existing work provides several mechanisms for such post-hoc validation, ranging from learned execution filters to formal proofs and end-to-end test suites. Some prior work on post-hoc verification arises in *program synthesis* and *code generation*, rather than translation, but the same mechanisms are transferable:

*8. Learning-Based Execution Filtering* Learned verifiers rerank or filter based on program execution outcomes (Ni et al., 2023). More formally: $V_\mathcal{C}(t, s) = 1$ iff the learned verifier, given $s$, $t$, and its execution result $R(t)$, classifies $t$ as satisfying $C(s)$.

*9. End-to-end test filtering.* Generated programs are executed against system- or repository-level test suites that exercise the entire program behavior, filtering or repairing candidates based on global pass/fail outcomes (Farrukh et al., 2025). More formally: $V_\mathcal{C}(t, s) = 1$ iff $t$ passes all tests in the full-suite $\mathcal{U}_s^{(e2e)}$, which covers cross-module and integration behaviors.

*10. Verified transpilation.* LLM-based translation can be paired with formal methods used to produce formal equivalence proofs, ensuring semantic preservation when proofs succeed (Bhatia et al., 2024). A key challenge is the scalability of these solvers. For large code bases, it is likely that compositional verification will be required in order for post-hoc verification of global equivalence to be proved, and LLMs may be required to produce artifacts for these proofs to be used in compositional reasoning in addition to the translated code.

| Approach | Top-level I/O | Internal | Beyond Functional | Domain | Representative Work |
|---|---|---|---|---|---|
| Learning-based execution filtering | ◖ | ○ | ○ | Code Generation | Ni et al. (2023) |
| End-to-end test filtering | ● | ◖ | ○ | Code Translation | Farrukh et al. (2025) |
| Verified transpilation | ● | ● | ●[1] | Code Translation | Bhatia et al. (2024) |

[1] Assume the property is explicitly modeled and compositionally enforced (incl. beyond-functional, e.g., constant-time, information-flow); otherwise mark as ◖.

Table 4: Coverage of verification-during-generation mechanisms across three correctness layers (top-level I/O, internal contracts, beyond-functional). ● means can guarantee; ◖ means contributes useful evidence; ○ means not helpful.

## B.3 COMPARATIVE IMPLICATIONS OF ONLINE VS. OFFLINE VERIFICATION

Our position is that any verification strategy must keep the translation+verification loop compositional: local obligations must accumulate into whole-program guarantees. We therefore compare

online and offline verification primarily through this lens, showing how they occupy different points on a spectrum of correctness-aware search, with distinct trade-offs in efficiency, flexibility, compositionality, alignment, and assurance.

- **Efficiency.** Online verification discards invalid prefixes early, pruning the search space before components are composed. This keeps the translation workflow incremental: modules that fail their local checks never pollute downstream reasoning. Offline verification defers filtering until the end, which can waste effort on invalid aggregates and forces coarse-grained rollback when a global check fails.
- **Flexibility.** Offline verification can mix and match verifiers (cheap first, expensive later), but without explicit decomposition it still evaluates monolithic artifacts. Online verification couples decoding with the verifier, reducing tool flexibility yet giving the verifier precise hooks at sub-component boundaries where local specs live.
- **Compositionality.** Online verification naturally enforces *local correctness*–syntax, typing, contracts–before assembly, preventing cascading errors and enabling modular proofs to accumulate. Offline verification must inspect whole programs unless the verifier itself is decomposed (e.g., assume-guarantee reasoning); otherwise it loses the very granularity compositional translation requires.
- **Alignment.** Embedding checks during generation steers the model toward producing spec-adherent pieces, reinforcing the habit of emitting components with explicit obligations. Offline verification is a pure filter; it cannot teach the model to respect compositional structure unless its signals are recycled into training.
- **Assurance.** Offline methods excel at heavyweight global obligations (symbolic execution, full-language equivalence) once the compositional scaffold is assembled. Online methods stay lightweight to preserve decoding throughput, but they guarantee that the artifacts handed to offline verifiers already satisfy the local contracts they rely on.

## USE OF LARGE LANGUAGE MODELS (LLMS)

Large Language Models were used only for two purposes: (i) to polish writing and improve the readability of the manuscript, and (ii) to assist with the retrieval and discovery of related work (e.g., helping to locate relevant papers). All conceptual ideas, experimental design, algorithm development, mathematical derivations, and data analysis were conceived, implemented, and verified by the authors. LLMs were not used to generate novel research content. Any automated assistance was carefully reviewed and edited by the authors to ensure accuracy and originality, in accordance with the ICLR 2026 guidelines on responsible LLM use.

