# OpenReview forum: "LLM-Based Code Translation Needs Formal Compositional Reasoning"
_ICLR.cc/2026/Conference — ICLR 2026 Conference Withdrawn Submission_

### Official Review · Reviewer_fbQF · 2025-10-31

**Soundness:** 1
**Presentation:** 2
**Contribution:** 1
**Rating:** 2
**Confidence:** 4

**Summary:**

This position paper argues that current evaluations of LLM-based code translation are overly reliant on tests, exact-match metrics, and syntactic similarity, while trustworthy translation requires formal, compositional correctness. The authors propose a layered notion of correctness that goes beyond passing I/O tests to include (1) top-level functional equivalence, (2) preservation of internal contracts and invariants (e.g., ownership, loop invariants), and (3) non-functional properties such as memory safety, security (e.g., constant-time behavior), and resource bounds. They conceptually claim that LLMs alone cannot ensure these obligations and advocate hybrid workflows where formal methods constrain and certify translations during and after generation, enabling scalable, component-wise verification across realistic codebases.

**Strengths:**

+ Code translation/migration is an important application (e.g., C to Rust migration) that LLMs can potentially help with, while having significant limitations for this task.
+ Equipping LLMs with formal compositional reasoning is indeed a promising direction, at a high level, to improve the performance and reliability for solving code translation tasks at a large scale.

**Weaknesses:**

**Positions are not new, and the high-level idea proposed by this work has been tried by existing works.**

The paper's core argument that testing alone cannot guarantee complete correctness while formal verification can provide such guarantees is well-established knowledge in the programming analysis and formal methods communities, rather than a surprising or novel position. More critically, the proposed approach of combining LLMs with compositional formal reasoning for code translation has already been actively explored in recent work, particularly in the C-to-Rust translation domain. For example, VERT (Yang et al., 2024) combines LLM-based translation with formal verification techniques including property-based testing and bounded model checking to verify equivalence between source and translated code. C2SaferRust (Nitin et al., 2025) adopts a neurosymbolic approach combining rule-based translation with LLM refinement and end-to-end test verification. I would suggest the authors to explicitly discuss its contributions relative to these concurrent efforts and expalin what unique technical or conceptual advances it offers beyond a general, well-received, high-level statement.

**Lack of in-depth discussion for the proposed solutions and the potential challenges.**

While the paper spent quite some space to criticizing test-based correctness (Section 2) and surveying existing verification mechanisms (Section 3 and Appendix B), it provides limited technical depth on what, concretely, the proposed LLM + compositional formal reasoning approach might look like. There is no detailed and concrete proposal for architectural and component designs, or implementation methodologies for audience/reviewers to assess how feasible the potential solution could be.

In addition, disappointingly, Section 4 identifies several fundamental open problems that are obviously critical and practical barriers to the formal-verification-based approach. I doubt that existing code translation works' focus on testing-based validation is due to these practical challenges, yet it provides no feasible solutions, proof-of-concept demonstrations, or detailed research roadmaps. For example, the paper acknowledges that "porting verification to code translation" faces unique challenges like incomplete verifier interfaces and cross-language semantic reasoning, but offers no technical proposals for overcoming these obstacles.

**Lack of empirical evidence (e.g., controlled experiments, apple-to-apple comparisons) to support its position.**

The paper makes several strong claims about the superiority of formal verification over testing and the necessity of compositional reasoning for scalable code translation, yet provides no controlled experiments or systematic empirical evaluation to substantiate these positions. The illustrative examples in the papers are just standalone cases demonstrating testing failures, but the paper does not quantify how frequently such failures occur in practice, nor does it provide comparative analysis showing that formal verification methods would reliably detect these issues at acceptable cost. Without controlled experiments demonstrating that the proposed formal verification pipeline achieves higher correctness rates, lower debugging costs, and better scalability than enhanced testing techniques, the paper's statements remain speculative and conceptual.

**Questions:**

- Given that recent work has already started implementing LLM-based code translation with formal verification and compositional reasoning, can you clarify what specific technical or conceptual advances your position paper proposes beyond these existing efforts?

- Section 4 identifies several critical open problems, but treats them as conceptual questions without feasibility analysis. Can you provide some preliminary evidence, even those from small-scale experiments, case studies, or literature analysis, that these challenges are feasible to be resolved? Otherwise, the proposed position seems weak if it is impractically difficult to be realised.

- The paper would be substantially strengthened by adding a few controlled experiments comparing test-based and verification-based approaches on the same benchmarks. Specifically, could you conduct an empirical study for that?

---

### Official Review · Reviewer_hC1v · 2025-11-01

**Soundness:** 3
**Presentation:** 3
**Contribution:** 2
**Rating:** 0
**Confidence:** 4

**Summary:**

This position paper argues that rigorous specification of correctness and formal verification are crucial in evaluating and ensuring the correctness of LLM-based translation.

**Strengths:**

The paper argues convincingly for the position.

**Weaknesses:**

- The paper has no theoretical contribution and a very limited experimental evaluation. It does not present a principled method for constructing correctness specifications, verifying them, and repairing the translation.
- Related to my point above, the paper is not a research paper but a position paper. In my opinion, the paper is not suitable to appear as a conference paper at ICLR. It is more suited to be submitted as a blog post: (https://iclr.cc/Conferences/2026/CallForBlogPosts), where "present[ing] novel perspectives or interpretations of existing machine learning concepts or techniques" is one of its interested area.

**Questions:**

N/A

---

### Official Review · Reviewer_znLJ · 2025-11-01

**Soundness:** 2
**Presentation:** 2
**Contribution:** 2
**Rating:** 2
**Confidence:** 5

**Summary:**

This position paper identifies a critical gap between superficial testing and the ideal equivalence metric for code translation. The authors propose a layered view of equivalence that encompasses 3 levels:

(1) top-level functional equivalence (input-output behavior)

(2) internal equivalence (contracts, invariants, and intermediate states)

(3) beyond-functional properties (memory safety, timing guarantees, resource bounds).

The central thesis is that decomposing verification into smaller, manageable proofs about components and their interactions is essential for scaling formal verification to realistic codebases. The paper advocates for a hybrid workflow where formal reasoning tools constrain, guide, and certify translation both during and after generation, transforming LLMs from statistical translators into reliable collaborators. The authors outline key open challenges, including cross-language reasoning, specification extraction, and verification of properties beyond functional equivalence.

**Strengths:**

## Novelty

1. **Valuable Conceptual Framework:** The proposed layered view of correctness (Section 2.3) is a valuable conceptual tool for thinking about and evaluating code translation. It provides a more nuanced and comprehensive understanding of what it means for a translation to be correct, encompassing not just functional equivalence but also internal consistency and non-functional properties. This framework could serve as a foundation for future benchmarks and evaluation methodologies, even if it requires further formalization.

2. **Taxonomy of Failure Modes:** The paper provides insightful examples of why test-based correctness fails (Section 2.1), identifying four key failure modes: (1) incomplete coverage (Figure 1), (2) inadequate context for deep specifications (Figure 2), (3) inability to capture challenging semantic requirements (Figures 3-4), and (4) backtracking in compositional translation (Reason 4). This taxonomy is well-illustrated with concrete examples and provides a solid foundation for understanding the limitations of current approaches.


## Significance

1. **Identifying a Critical Problem:** The paper identifies a problem of critical importance. As LLMs are increasingly used for code generation and translation, ensuring the correctness of the generated code is paramount, especially in safety-critical domains. This paper makes a strong case for why this is a hard problem and why a new approach is needed. The potential impact on domains like cryptographic implementations, operating systems, and embedded systems is substantial.

**Weaknesses:**

## Novelty

1. **Missing Formal Definition of Cross-Language Equivalence:** While program equivalence within a single language has a solid theoretical foundation in programming language theory, the paper does not provide a formal definition of what it means for a program in one language to be equivalent to a program in another language.

    Specifically, **some programs in language A can’t be translated as-is to language B**. Languages like JavaScript or Python, and even C/C++ when they use unchecked casts or layout tricks, allow programs whose behavior depends on runtime or representation details that a strongly typed target refuses to express. Those programs can’t be translated as a total, type-preserving, semantics-preserving translation.

    This is the core theoretical contribution that such a paper should make. The layered correctness framework is presented conceptually, but without formal semantics or a rigorous definition of equivalence across languages with different type systems, memory models, and operational semantics. It would significantly strengthen the paper to develop a formal theory of cross-language equivalence, perhaps using relational semantics or a common intermediate representation.

2. **Limited Conceptual Novelty in Formal Methods:** The core ideas of compositional verification [2], Hoare logic [3], and contract-based design [4] are well-established in the formal methods community. While the paper provides a valuable synthesis and application of these concepts to LLM-based code translation, it does not introduce fundamentally new verification techniques or theoretical frameworks. The novelty lies primarily in the problem formulation and the vision for combining LLMs with existing formal methods, rather than in the development of new formal reasoning principles. It would strengthen the paper to more explicitly acknowledge which aspects of the proposed approach are novel versus which are adaptations of existing techniques, and to provide a more detailed comparison with recent work on verified code translation [5, 6, 7].


## Soundness

1. **Absence of Empirical Evidence to Support Central Claims:** The paper makes strong claims about the inadequacy of test-based evaluation, but provides no empirical data or experimental results to support these claims. While the illustrative examples in Figures 1-4 are helpful, they are anecdotal and not a substitute for systematic, quantitative analysis. The paper would be significantly strengthened by conducting a medium-scale empirical study on popular code translation benchmarks. Such a study could involve manually analyzing a sample of problems to determine whether the provided test cases are sufficient to capture the full semantics of the programs, quantifying the percentage of problems where test cases are weak or incomplete, and using mutation testing to generate semantically incorrect variants that still pass the test suite. This would transform the paper's central claim from a plausible assertion into a well-supported empirical finding.

2. **The Unaddressed Problem of Specification Definition:** The paper advocates for formal specifications but does not address the fundamental question: **what kind of formal specification should we have for code translation?** The appendix and Table 1 provide a catalog of verification techniques (grammar-constrained decoding, type checking, contract verification, etc.), but they treat the specification as a given. This leads to circular reasoning: the paper argues we need formal specifications to verify correctness, but does not provide a principled methodology for defining what those specifications should be for a given translation task. It would strengthen the paper to include a section on specification engineering for code translation, discussing different approaches for obtaining specifications (manual specification, inference from documentation, synthesis from examples), analyzing trade-offs between different types of specifications (lightweight contracts vs. full functional specifications), and proposing a methodology for determining what should be specified for a given translation context.


## Effectiveness

1. **Unvalidated Effectiveness Claims:** The effectiveness of the proposed hybrid workflow cannot be evaluated as it has not been implemented or tested. The paper presents a compelling vision, but the proof of its effectiveness will depend on successful implementation and empirical validation. The paper would be strengthened by at least a preliminary case study or a small-scale experiment demonstrating the feasibility of the approach, even if it is limited in scope.

2. **Unclear Cost-Benefit Trade-Off:** The paper does not discuss the computational and human costs of the proposed approach. Formal verification is notoriously expensive, both in terms of computation time and human effort. The paper should provide at least a rough estimate of the expected costs and benefits, and discuss scenarios where the approach would be cost-effective versus scenarios where it would not be. This would help readers understand the practical viability of the proposed approach and make informed decisions about when to apply it.


## References

[1] [CodeBLEU: a Method for Automatic Evaluation of Code Synthesis (Ren et al., 2020)](https://arxiv.org/abs/2009.10297)

[2] [Circular Compositional Reasoning about Liveness (McMillan, 1999)](https://link.springer.com/chapter/10.1007/3-540-48153-2_22)

[3] [An Axiomatic Basis for Computer Programming (Hoare, 1969)](https://dl.acm.org/doi/10.1145/363235.363259)

[4] [Design by Contract](https://en.wikipedia.org/wiki/Design_by_contract)

[5] [Verified Code Transpilation with LLMs (Bhatia et al., 2024)](https://arxiv.org/abs/2406.03003)

[6] [Polyver: A Compositional Approach for Polyglot System Modeling and Verification (Chen et al., 2025)](https://arxiv.org/abs/2503.03207)

[7] [Syzygy: Dual Coderust-to-Safe-Rust Translation Using LLMs and Dynamic Analysis (Shetty et al., 2024)](https://arxiv.org/abs/2412.14234)

**Questions:**

## Clarification Questions

1. The paper proposes a hybrid workflow that combines LLMs and formal verification tools (Section 3, Appendix B). Could the authors elaborate on how they envision the interaction between these two components? Specifically, how would the feedback from the verification tool be used to guide the LLM in generating a correct translation? Would the LLM be fine-tuned on verification failures, or would the feedback be incorporated into the prompt at inference time? What is the expected number of iterations before convergence to a correct translation?

2. The paper mentions the challenge of specification extraction (Section 4). Could the authors discuss any promising approaches for automatically or semi-automatically extracting formal specifications from informal documentation or existing code? Are there specific types of specifications (e.g., functional contracts, security properties, resource bounds) that are more amenable to automatic extraction than others? How would the authors handle cases where the documentation is incomplete, ambiguous, or inconsistent with the actual code behavior?

3. In Section 2.3, the paper discusses "beyond functional equivalence" properties such as memory safety and timing guarantees. How would the proposed approach handle properties that are inherently language-specific or platform-specific? For example, Rust's ownership system provides memory safety guarantees that are not present in C. Would the translation process need to insert additional runtime checks or restructure the code to preserve these properties? How would the authors handle cases where the source and target languages have fundamentally different memory models or concurrency semantics?

4. Table 1 categorizes different verification approaches by when they are applied (during vs. after generation) and what properties they can verify. Could the authors clarify what they mean by "can guarantee" (●) versus "contributes useful evidence" (◐)? What level of assurance is required for a property to be considered "guaranteed"? Is this a formal proof, or a high-confidence probabilistic guarantee?


## Discussion Questions

1. The paper identifies several cases where test-based evaluation fails, but does not discuss cases where formal verification might also fail or be insufficient. Could the authors discuss the limitations of formal verification in the context of code translation? For example, formal verification typically requires complete and correct specifications, but in practice, specifications are often incomplete, incorrect, or ambiguous. How would the proposed approach handle cases where the specification itself is the source of the problem?

2. The paper argues that compositional reasoning is essential for scalability, but compositional verification requires well-defined module boundaries and interfaces. How would the proposed approach handle legacy codebases with poor modularity or unclear interfaces? Would the system need to first refactor the code to make it more amenable to compositional verification? How would the authors ensure that such refactoring preserves the original semantics?

3. The paper focuses on source-to-source code translation, but many real-world translation tasks involve not just the code itself but also the surrounding ecosystem (build systems, dependencies, configuration files, documentation, etc.). How would the proposed approach extend to these broader translation scenarios? Would the formal specifications need to cover these aspects as well? How would the authors handle cases where the source and target ecosystems are fundamentally different (e.g., translating a Python project with pip dependencies to a Rust project with cargo dependencies)?

---

### Official Review · Reviewer_WRoa · 2025-11-03

**Soundness:** 2
**Presentation:** 2
**Contribution:** 2
**Rating:** 4
**Confidence:** 3

**Summary:**

This paper argues that correctness should be the guiding principle of LLM-based code translation. It highlights the inadequacy of current test-based evaluation and proposes that formal and compositional reasoning are essential for achieving trustworthy translation. The authors advocate for layered correctness definitions (functional, internal, and beyond-functional). The paper positions itself as a call to integrate formal methods into the translation pipeline and outlines several open challenges.

**Strengths:**

* I agree with the overall sentiment of the paper that code translation seems to be heading into a direction where scalability is the main goal, with correctness secondary, and the fact that this needs to change if we are to have good, reliable translation techniques.

* The paper articulates known concerns in the code-generation community, namely that test-based evaluation fails to capture correctness, and provides a coherent perspective.

* The layered notion of correctness (top-level I/O, internal contracts, and beyond-functional properties) provides a useful taxonomy for reasoning about translation quality.

**Weaknesses:**

* While its arguments are well motivated, the paper does not present a new technique, experimental framework, or formal definition that make them operational. I think that the fact that formal methods are not used more in code translation is ultimately due to limitations in existing verification techniques (with some issues discussed in the open problems section): inferring formal specs (from code or from from semi-structured text) is extremely hard, especially for real-world codebases; specifications are often highly domain-specific; verification tools are specialised to a particular language; verification techniques have reduced scalability. I don't see an immediate solution to any of these problems.

* I think that most of the suggested techniques have been attempted in some manner, as explained below. Therefore, this work reads at times as a synthesis of existing directions rather than a new agenda that pushes the conceptual or methodological frontier of code translation.

-- Formal verification: [2] uses bounded model checking to check the semantic equivalence of the translation and the source although it only works on small codes.

-- Compositional translation and validation: [3] performs a modular two-tiered generation and validation, where the first tier makes use of type checks to validate the translation, and the second uses I/O validation. Additionally, it introduces a notion of "type compatibility", which maintains structs exactly, essentially enforcing internal equivalence as recommended in the current submission.

-- Using the LLM to infer specifications: [1] attempts to use LLMs for generating different types of specifications. These are not currently used for verification, but only to guide the translation.

* The examples (Figures 1–5) are useful but mostly reproduce standard translation pitfalls rather than motivating a distinct formal framework.

[1] Yang et al, VERT: Verified Equivalent Rust Transpilation with Few-Shot Learning.

[2] Nitin and Ray, "SPECTRA: Enhancing the Code Translation Ability of Language Models by Generating Multi-Modal Specifications"

[3] Zhang et al, "Scalable, Validated Code Translation of Entire Projects using Large Language Models", PLDI'25

**Questions:**

Is there a realistic path toward coupling current LLMs with formal verifiers at scale, given the limitations of specification inference and verification for real-world codebases?

---

### Note · Authors · 2025-11-20

I have read and agree with the venue's withdrawal policy on behalf of myself and my co-authors.